# Astrocytes-Derived Small Extracellular Vesicles Hinder Glioma Growth

**DOI:** 10.3390/biomedicines10112952

**Published:** 2022-11-17

**Authors:** Carmela Serpe, Antonio Michelucci, Lucia Monaco, Arianna Rinaldi, Mariassunta De Luca, Pietro Familiari, Michela Relucenti, Erika Di Pietro, Maria Amalia Di Castro, Igea D’Agnano, Luigi Catacuzzeno, Cristina Limatola, Myriam Catalano

**Affiliations:** 1Department of Physiology and Pharmacology, Sapienza University, 00185 Rome, Italy; 2Department of Chemistry, Biology and Biotechnology, University of Perugia, 06123 Perugia, Italy; 3Division of Neurosurgery, Department of Human Neurosciences, Policlinico Umberto I, Sapienza University of Rome, 00185 Rome, Italy; 4Department of Anatomical, Histological, Forensic Medicine and Orthopedics Sciences, Sapienza University, 00185 Rome, Italy; 5Institute of Biomedical Technologies, CNR, 20054 Segrate, Italy; 6Department of Physiology and Pharmacology, Laboratory Affiliated to Istituto Pasteur Italia Fondazione Cenci Bolognetti, Sapienza University, 00185 Rome, Italy

**Keywords:** small extracellular vesicles, astrocytes, volume-regulated anion channels, miR124, glioma

## Abstract

All cells are capable of secreting extracellular vesicles (EVs), which are not a means to eliminate unneeded cellular compounds but represent a process to exchange material (nucleic acids, lipids and proteins) between different cells. This also happens in the brain, where EVs permit the crosstalk between neuronal and non-neuronal cells, functional to homeostatic processes or cellular responses to pathological stimuli. In brain tumors, EVs are responsible for the bidirectional crosstalk between glioblastoma cells and healthy cells, and among them, astrocytes, that assume a pro-tumoral or antitumoral role depending on the stage of the tumor progression. In this work, we show that astrocyte-derived small EVs (sEVs) exert a defensive mechanism against tumor cell growth and invasion. The effect is mediated by astrocyte-derived EVs (ADEVs) through the transfer to tumor cells of factors that hinder glioma growth. We identified one of these factors, enriched in ADEVs, that is miR124. It reduced both the expression and function of the volume-regulated anion channel (VRAC), that, in turn, decreased the cell migration and invasion of murine glioma GL261 cells.

## 1. Introduction

Astrocytes have a substantial function in maintaining brain homeostasis. These cells control neurotransmitter concentrations in the synaptic cleft to support physiological neuronal activity and are involved in the formation and function of the blood brain barrier (BBB) [1]. Astrocytes can also respond to harmful stimuli with transcriptional changes, as in the process of astrogliosis, that depend on the intensity and duration of the damage. Reactive astrocytes can support brain regeneration in many cerebral pathologies [2,3,4]. The harmful/protective astrocyte function is mediated by the release of soluble factors (cytokines, neurotrophic and growth factors) [5], by cell-cell contact in the participation to the BBB [4] or by astrocyte-derived extracellular vesicles (ADEVs) that mediate astrocyte paracrine signaling and permit synaptic function [6,7].

The most common primary brain tumors in both adults and children are gliomas whose most aggressive form is Glioblastoma (GBM). Even if new therapeutic strategies are constantly being examined, GBM is associated with high morbidity and mortality and remains incurable [8]. The tumoral mass is composed of heterogeneous cell types that have a high rate of invasion and proliferation, making any therapeutic approach difficult. Parenchymal and stromal cells that represent the tumor microenvironment (TME), under the influence of the tumor cells, lose their homeostatic activities. Tumor cells take advantage of the TME to grow and to invade [9]. The TME of GBM is composed by microglia, infiltrating immune cells, neural precursor cells, vascular cells and astrocytes [10].

In this context, astrocytes represent a first defense against tumor invasions as the source of a plasminogen activator that catalyzes neuron-derived plasminogen in plasmin, which in turn induces the release of FasL from astrocytes. Fas is expressed by tumor cells and activates proapoptotic caspases [11]. Astrocytes influenced by cancer cells, promote glioma progression; this role depends on the intrinsic heterogeneity of astrocytes, on astrocyte-GBM cell-to-cell contacts [12] and also on ADEVs. Specifically, glioma-influenced ADEVs translocate PTEN-targeting miRNAs guilty to increase proliferation and to reduce apoptosis of tumor cells [13].

Among many malignant characteristics of GBM cells, there are uncontrolled proliferation and high rate of invasiveness that make the tumor difficult to treat both pharmacologically and surgically. GBM cells display a higher survival during exposure to insults (such as hypoxic conditions) compared to normal non-malignant cells [14]. The regulatory volume decrease (RVD) is a complex homeostatic process displayed by virtually all cells, that allows them to recover the original volume lost in response to osmotic stress. RVD is mainly modulated by the concerted activity of Cl^-^ and K^+^ channels that mediate a net efflux of KCl from the cell, followed by obligatory osmotic water, with the consequent recovery of the original cell volume. Several brain pathologies (i.e., hyponatremia and epilepsy) present a defective control of cell volume and consequently of the RVD [15,16,17]. One of the main players in the modulation of RVD is the volume-regulated anion channel (VRAC) which mediates the swelling-activated Cl^-^ current (I_Cl,swell_), a ubiquitous current activated by hypotonic stress [18,19]. GBM cell invasion largely depends on the activity of ion channels [20,21], included VRACs. These channels are highly expressed in GBM cells [22] and are heteromeric channels made of the protein leucine-rich repeat containing 8A (LRRC8A) and at least one other LRRC8 isoform among LRRC8B, LRRC8C, LRRC8D and LRRC8E [23].

Gene expression is finely regulated by small non-coding RNAs called microRNAs (miRNAs) [24]. miRNAs are short single-stranded non-coding RNA molecules made of about 21–23 nucleotides that post-transcriptionally regulate gene expression [25]. In addition, some nucleus-located miRNAs are found to activate or silence the transcription of target genes [26].

Among others, miR124 is highly expressed in the central nervous system, and is downregulated or completely absent in high-grade gliomas [27]; its absence correlates with an increased invasive capacity of tumor cells [28]. Moreover, miR124 is conveyed by microglia-released sEV to glioma cells, contributing to reduce the neurotoxic release of glutamate [29].

Herein, we aimed to study the role of healthy ADEVs (i.e., uninfluenced by tumor cells) in the progression of GBM to better understand the pathophysiology and hypothesize that these vesicles can be useful as a first line of defense for therapeutic purposes. The results obtained in this study show that healthy ADEVs, through the specific action of miR124, oppose in vitro and in vivo GBM growth and invasion, by reducing the expression of VRAC-mediated I_Cl,swell_ on glioma GL261 cells, a mechanism by which healthy astrocytes try to limit tumor progression.

## 2. Materials and Methods

**Cell lines**. GL261 murine glioma cells were cultured in DMEM (Thermo Fisher Scientific, Waltham, MA, USA), supplemented with 20% heat-inactivated exosome-depleted FBS (Thermo Fisher Scientific, Waltham, MA, USA), 100 IU/mL penicillin G (Thermo Fisher Scientific, Waltham, MA, USA), 100 μg/mL streptomycin (Thermo Fisher Scientific, Waltham, MA USA), 2.5 μg/mL amphotericin B (Thermo Fisher Scientific, Waltham, MA, USA) and grown at 37 °C in a 5% CO2 and humidified atmosphere.

**Primary astrocyte cultures**. Astrocyte cells were obtained from mixed glia cultures derived from the cerebral cortices of post-natal day 0–2 (P0–P2) C57BL6/N mice. Cortices were chopped and digested in 15 U/mL papain (Merck KGaA, Darmstadt, Germany) for 20 min at 37 °C. Cell suspensions were plated (5 × 10^5^ cells/cm^2^) on poly-L-lysine hydrobromide (Merck KGaA, Darmstadt, Germany) (0.1 mg/mL) coated flasks in a growth medium supplemented with 10% heat-inactivated exosome-depleted FBS. After 9–11 days, cultures were shaken for 2 h at 37 °C to detach microglia cells and to obtain a pure primary murine astrocytes cell culture.

**Cell transfection**. GL261 cells (2 × 10^5^ cells/mL) plated onto 24-well plates were mock-transfected or transfected with a miR-124 mimic sequence (Thermo Fisher Scientific, Waltham, MA, USA) via Lipofectamine 3000 Reagent (Thermo Fisher Scientific, Waltham, MA, USA) in Opti-MEM (Thermo Fisher Scientific, Waltham, MA, USA) according to manufacturer instructions. All types of analysis were performed 48 h after transfection.

**Vector construct and luciferase assay**. The mRNA sequence of murine LRRC8C was retrieved from GenBank (NM_133897.2). Based on this sequence, oligonucleotide couples flanking the putative region binding miR-124 (according to Tarbase, position 4035–4041 of LRRC8C 3′-UTR) were designed, synthesized and used as primers for q-PCR (forward primer: AACATTGGACAGCTTTTATTGACC; reverse primer: ACACCACACAGCGAGATAACA). The PCR product was cloned into a luciferase reporter plasmid, pmiR-report (Thermo Fisher Scientific, Waltham, MA, USA) for luciferase assay. To assess luciferase activity, GL261 cells were plated onto 24-well plates (2 × 10^5^ cells/mL) and transfected with 150 μg of the pmiR-construct, 50 nM of the miR-124 mimic and renilla luciferase plasmid (as internal transfection control) via Lipofectamine (as stated above). After 24 h, the medium was discarded, and the cells were collected and homogenized in 70 μL of lysis buffer. Cell lysates were centrifuged for 5 min at 12,000× *g*. Luciferase activity was detected using the Dual-Luciferase Reporter Assay System (Promega, Milan, Italy) and measured by the Gloxmax 96 luminometer (Molecular Devices, San Jose, CA, USA). For each sample, firefly luciferase activity was normalized to renilla luciferase activity.

**Extraction of ADEVs**. Astrocyte supernatant was collected and centrifuged at 800 g for 5 min to remove cell debris. The supernatant was centrifuged at 10,000× *g* for 30 min at 4 °C and then to ultracentrifugation at 100,000× *g* for 1 h at 4 °C. The resulting pellet, containing sEVs, was re-suspended in a 0.22 μm filtered PBS (Thermo Fisher Scientific, Waltham, MA, USA) or in a sEV-free fraction medium depending on the subsequent experiments. To obtain a sEV-free fraction medium, the media were submitted to the above reported steps of centrifugation and supernatants were collected and used as vehicles.

**Nanoparticle tracking analysis (NTA)**. ADEV samples obtained from the 100,000× *g* centrifugation step were resuspended in PBS and analyzed using Nanosight NS300 (Malvern Panalytical, Malvern, UK). Videos were analyzed by the inbuilt NanoSight Software NTA 3.4 Dev Build 3.4.4. The camera type, camera level and detection threshold were sCMOS, 14 and 4, respectively. The number of completed tracks in NTA measurements was 5 (a 60 s movie was registered for each measurement). The sample was diluted in a 0.22 µm filtered PBS to a final volume of 1 mL. The ideal concentration was assessed by pre-testing the optimal particle per frame value (20–100 particles per frame).

**Western blot Analysis**. ADEVs, astrocytes and GL261 cells were lysated in a RIPA buffer (50 mM TrisHCl, 150 mM NaCl, 1% Triton, 0.1% SDS, 1% deoxycholate, 2 mM EDTA, pH 7.5) added with a Protease and Phosphatase Inhibitor Cocktail (Sigma Aldrich, Milan, Italy). Protein samples were separated by 10% SDS-polyacrylamide gel electrophoresis (20 µg/lane). The blot has been cut probing different regions of the same blot with the following primary antibodies: CD81 (Cell Signaling, Danvers, MA, USA), 1 : 500, GAPDH (Merck, Darmstadt, Germany) 1:500, LRRC8C (Proteintech, Rosemont, IL, USA) 1:1000, actin (Sigma Aldrich, Milan, Italy) 1:500; HRP-tagged goat anti-rabbit or anti-mouse IgG were used as secondary antibodies (Dako, Cernusco sul Naviglio, Italy) 1:1000;; the detection was performed through the chemiluminescence assay with an Amersham ECL Prime Western Blotting Detection reagent (Cytiva, Little Chalfont Buckinghamshire, UK). Densitometric detection was carried out with the Quantity One software (version 4.6.6, Bio-Rad, Hercules, CA, USA).

**Transmission electron microscopy (TEM**). ADEVs were resuspended in 2.5% glutaraldehyde in phosphate buffer 0.1 M at pH 7.4. After fixation, samples were rinsed in phosphate buffer 0.1 M at pH 7.4, post-fixed with 1% osmium tetroxide (OsO_4_) (Agar Scientific, Stansted, UK) in phosphate buffer 0.1 M at pH 7.4 and rinsed with ultrapure water (Direct Q^®^ 3UV, Merck, Milan, Italy) for 20 min. To obtain an increased contrast and a sharp delineation of plasma membrane-bound sEVs, samples were stained for 2 min with Uranyless© (EM stain solution, Electron Microscopy Sciences, Hatfield, PA, USA) and washed for 10 sec in ultrapure water (Direct Q^®^ 3UV, Merck, Milan, Italy). Observations were carried out by TEM (Carl Zeiss EM10, Thornwood, NY, USA), operating at 60/80 kV conditions and equipped with a digital camera (AMT CCD, Deben XR-80, 8 Mp, UK Ltd., Suffolk IP309QS, UK)

**MTT-based viability assay**. GL261 cells were seeded into 24-well plates (1 × 10^4^ cells/well) and treated with a vehicle (sEV-free fraction medium), with ADEVs isolated from healthy astrocytes or with ADEVs derived from astrocytes stimulated with a glioma (GL261) conditioned medium (GCM). The ratio ADEVs donor cells and GL261 was 5:1. MTT (500 μg/mL) was added into each well for 1.5 h. DMSO was then added to stop the reaction and the formazan produced was measured at 570 nm. Viability of cells was expressed relative to absorbance.

**Tumor cell implantation and mouse treatment**. Experiments were approved by the Italian Ministry of Health (protocol number 694/2016- PR) in accordance with the ethical guidelines on the use of animals from the EC Council Directive 2010/63/EU. GL261 cells (1 × 10^5^ in 4 µL PBS) were injected in the right striatal brain region of eight-week-old male C57BL/6N mice. During surgery, a guide cannula was placed 2 mm deep in the striatum and it was fixed with quick-setting cement. sEVs obtained from 10^7^ primary astrocytes were re-suspended in 4 μL of PBS and infused via cannula two times: 7 and 14 days after the tumor implantation. The day after the second infusion, animals were sacrificed and analyzed for tumor size or for immunofluorescence. 

**Tumor volume analysis**. After 15 days of the GL261 injection, animals were killed and the brains were isolated. Tumor volume was evaluated with a hematoxylin–eosin staining standard protocol. After staining, brain slices (20 μm of thickness) were analyzed by the Image Tool 3.0 software (University of Texas, Health Science Center, San Antonio, TX, USA). To measure the tumor area, volume was calculated according to the formula (volume = t × ΣA), where A = tumor area/slice and t = thickness.

**Immunofluorescence**. Coronal brain sections (20 μm) were washed in PBS, blocked (3% goat serum in 0.3% Triton X-100) for 1 h at RT and incubated with rabbit anti-Ki-67 (1:50, Spring Bioscience, London, UK) or rabbit anti-LRRC8C (1:80, Proteintech, Rosemont, IL, USA). The slices were washed in PBS and stained at RT for 1 h with the secondary antibody conjugated to Alexa Fluor 568 (1:500, Thermo Fisher Scientific, Waltham, MA USA). Hoechst 33342 (stained at RT for 1 h; 1:1000 Thermo Fisher Scientific, Waltham, MA USA) was used for nuclei visualization. The slices were mounted with a Dako Fluorescence Mounting Medium (Agilent, Santa Clara, CA USA). Analysis was performed using a fluorescence microscope (Nikon, Leuven, Belgium) by MetaMorph 7.6.5.0, measuring the signal coverage area of Ki-67 normalized by tumor area or LRRC8C normalized by area, upon setting the scale of threshold.

**Wound Healing**. GL261 cells (5 × 10^5^/mL) were seeded into a cell culture insert (Ibidi, Gräfelfing, Germay) placed in a Petri Dish. Once attached to the substratum, the inserts were removed, leaving a central 500 µm cell-free septum in which cells could migrate. A cell medium with or without ADEVs (ratio 5:1 = donor:target cells) was added. Cells were incubated with a cell cycle blocker (AraC, 10 µM; Thermo Fisher Scientific, Waltham, MA USA) to prevent GL261 proliferation for the time of the experiment. Dishes were maintained at 37 °C, 5% CO_2_. Pictures of time 0 (0 h) and 24 and 48 h after treatment were taken at a phase contrast microscope (Nikon, Leuven, Belgium) and processed through the MetaMorph 7.6.5.0 software (Molecular Devices, San Jose, CA, USA). GL261 migration was evaluated by the area between the two cell fronts (by ImageJ Software (v.1.53e, NIH, USA) and data are expressed as a % of area occupied by cells.

**Invasion Assay**. GL261 cells were plated on Matrigel-coated transwells (BD-Falcon, Milan, Italy) in the presence or absence of ADEVs. Pro-invasive stimulus 10% FBS was added in the lower chamber. Cells were incubated 24 h at 37 °C and then fixed (with ice-cold 10% trichloroacetic acid, 10 min) and stained (with a solution containing 50% isopropanol, 1% formic acid and 0.5% brilliant blue R 250). Stained cells were counted with a × 40 objective.

**Cell volume measurements**. Cells were perfused with an extracellular Ringer’s solution consisting of NaCl 140 mM, KCl 5 mM, CaCl_2_ 2 mM, MgCl_2_ 2 mM, MOPS 5 mM and glucose 10 mM (pH 7.40). For the study of the hypotonic-induced cell swelling effect, a 30% hypotonic solution was applied. The telative projection area of cells was measured using a video-imaging technique. The cells were observed through a 40× objective lens of a microscope connected to a video camera (Axiocam/Cm1, Zeiss, Dublin, CA, USA). Images of the cells were saved as TIFF files and the area of each image was subsequently measured using ImageJ. Images were acquired every 2 min for the experiments shown in Figure 1. The relative cell area (Arel) was calculated by dividing the area of the cell at each time point (At) upon exposure to the hypotonic solution by the initial area (A0) taken under isotonic conditions (more precisely, the average of the three time points recorded in the first 4 min). The extent of RVD was measured 60 min after the onset of the regulatory response, which started soon after the cell reached the peak of swelling under the hypotonic solution and is calculated as follows: %RVD = ((Arel,peak−Arel,60 min)/(Arel,peak))*100, where Arel,peak and Arel,60 min are the relative areas at the peak of cell swelling and at the end of the experiment, respectively.

**Electrophysiological recordings**. The whole-cell dialyzed configuration was used for electrophysiological recordings from GL261 cells. Currents and voltages were amplified with a HEKA EPC-10 amplifier (List Medical, Darmstadt, Germany), digitized with a 12-bit A/D converter (TL-1, DMA interface; Axon Instruments, Foster City, CA, USA) and analyzed with the software Patch-Master package (version 2_60, ELEKTRONIK) and Microcal Origin 8.0. For on-line data collection, macroscopic currents were filtered at 3 kHz and sampled at 200 ms/point. The external standard solution contained: NaCl 140 mM, KCl 5 mM, CaCl_2_ 2 mM, MgCl_2_ 2 mM, MOPS 5 mM and glucose 10 mM, (pH 7.40). The internal solution contained: KCl 155 mM, EGTA-K 1 mM, MOPS 5 mM and MgCl_2_ 1 mM (pH 7.20). Access resistances ranged between 8 and 15 MΩ and were actively compensated to ~50%. 4-[(2-Butyl-6,7-dichloro-2-cyclopentyl-2,3-dihydro-1-oxo-1H-inden-5-yl)oxy] butanoic acid (DCPIB)(Tocris Bioscience, Bristol, UK), a potent and recognized inhibitor of I_Cl,swell_ mediated by VRAC, was dissolved in dimethyl sulfoxide (DMSO, Sigma-Aldrich, Milan Italy) at the stock concentration of 10 mM and used at the final concentration of 10 µM in the recording solution. This concentration completely blocks the IC_l,swell_ mediated by VRAC as previously reported [13]. The highest DMSO concentration in the recording solutions was 0.1%. To activate the I_Cl,swell_ through hypotonicity, we used the 30% hypotonic solution (Hypo). To eliminate the contribution of both voltage- and Ca^2+^-activated K^+^ currents, estimates of I_Cl,swell_ were taken at −80 mV, the equilibrium potential for K+, in the presence of 3 mM extracellular TEA, a blocker of large-conductance Ca^2+^-activated K^+^ channels (BK). All reagents were fresh and daily solubilized at the concentrations stated, and the bath applied with a gravity perfusion system. Experiments were carried out at room temperature (18–22 °C).

**Real Time PCR (qPCR)**. Total RNAs were extracted from cells or from ADEVs with a Trizol reagent (Sigma Aldrich, Milan, Italy) or the Total EXO RNA Extraction Kit (Sigma Aldrich, Milan, Italy) according to the manufacturer’s description. RNAs extracted from all samples were quantified and retro-transcribed using iScript Reverse Transcription Supermix (Bio-Rad, Hercules, California, US). Real time PCR (qPCR) was carried out in an I-Cycler IQ Multicolor RT- PCR Detection System (Bio-Rad, Hercules, California, US) using the SsoAdvanced Universal SYBR Green Supermix (Bio-Rad, Hercules, California, US). The PCR protocol consisted of 40 cycles of denaturation at 95 °C for 30 s and annealing/extension at 58 °C for 30 s. The Ct values from each gene were normalized to the Ct value of *gapdh*. MicroRNAs levels were measured with the TaqMan MicroRNA Assay kit (Thermo Fisher Scientific, Waltham, MA USA)) according to the manufacturer’s protocol. Equal amounts of RNAs of all samples (cells and sEVs) were analyzed. For quantification analysis, the comparative threshold cycle (Ct) method was used. The Ct values of each gene were normalized to the Ct value of U6 in the same RNA sample (ΔCt). Relative quantification was performed using the 2-ΔΔCt method and expressed as fold increase. Primer sequences: *gapdh*, forward: TCGTCCCGTAGACAAAATGG, reverse: TTGAGGTCAATGAAGGGGTC; LRRC8A, forward: TACCTGGACCTCAGCCACAACA, reverse: CTTCCGACACTGGAAGAGCTCT LRRC8C, forward: GTGTTCACGGATTACCTCTCGG, reverse: ACGGAAGAGTGGTTCTGAGCAG.

**Statistical analysis**. Data are expressed as means ± SEM. Student’s *t*-test, paired *t*-test and one-way or two-way analysis of variance (ANOVA) were performed. A value of *p* ≤ 0.05 was considered significant. Statistical analyses were performed using the Prism GraphPad Software (v9.3.1, San Diego, CA, USA).

## 3. Results

### 3.1. Astrocyte-Derived sEVs (ADEVs) Limit Glioma Growth

Primary murine astrocytes were used as the source of sEVs. Astrocyte-derived sEVs (ADEVs) were characterized for dimension, exosomal marker and morphology. Analysis by NTA to define sEV size and number showed that the mean size of ADEVs was below 200 nm (162.6 ± 1 nm; N = 5) and that the sEV mean concentration was 4.2 × 10^7^ (±2.6 × 10^3^) nanoparticles/mL (Figure 1a). Western blot analysis revealed that the tetraspanin exosomal marker CD81 was enriched in ADEVs vs donor cells; the enzyme GAPDH was used as a negative control (Figure 1b). Transmission electron microscopy (TEM) was performed on ADEVs and a cup-shaped morphology was observed (Figure 1c). Of note, the single vesicle measured about 70 nanometers but sEVs tended to cluster together, mainly in groups of two or three vesicles, compatible with the NTA measurements.

To investigate a possible antitumoral effect of ADEVs, we performed in vitro tests to assess their capacity to modulate glioma cell viability. We treated the murine glioma cell line GL261 with ADEVs and analyzed cell viability by the MTT assay. The ratio between ADEVs donor cells and GL261 used was 5:1 because: (i) the ratio 1:1 was not efficacious to reduce GL261 viability at any time point analyzed; (ii) the ratio 10:1 was significantly different vs vehicle-stimulated cells (Student’s *t*-test revealed: at 24 h, vehicle vs 10:1 *p* = 0.003; at 48 h, vehicle vs 10:1 *p* = 0.029; at 72 h, vehicle vs 10:1 *p* = 0.039) but was not significantly different than the ratio 5:1 (data not shown). ADEV administration reduced GL261 cell viability after 24, 48 and 72 h of incubation (Figure 1d). In contrast, ADEVs isolated from astrocytes treated with the glioma-conditioned medium (GCM) increased GL261 cell viability, indicating that only sEVs released by healthy astrocytes have an antitumoral effect (Appendix A).

This result prompted us to further investigate the effect of ADEVs in a mouse model of glioma. We implanted GL261 glioma cells into the brain (striatal region) of adult male mice and treated them with ADEVs 7 and 14 days after tumor implantation. As a control, glioma-bearing mice were treated as described with a sEV-free fraction medium (vehicle). A group of animals was treated with GCM-ADEVs. Mice were sacrificed 17 days after the implant and we observed both a reduction of the tumor volume (Figure 1e) and a reduction of tumor cell proliferation, measured as Ki67+ cells in the tumor core (Figure 1f,g). The tumor volume in GCM-ADEVs-treated mice was increased with respect to the vehicle-treated mice (Appendix A). The ADEVs treatment significantly prolonged the survival of glioma-bearing mice (Figure 1h), confirming the data obtained in vitro. 

### 3.2. ADEVs Limit Glioma Cell Invasion 

To test the anti-tumoral efficacy of ADEVs as an ability to reduce glioma cell migration, we analysed their effect on glioma cell motility by a wound healing assay and observed a reduction in wound closure upon 24 and 48 h of ADEV treatment (Figure 2a; representative images are reported in Figure 2b). We further examined the invasion ability of ADEVs-treated glioma cells by performing a Matrigel invasion assay and observed that ADEV treatment blocked the FBS-induced tumor cell invasion (Figure 2c). These results indicate that ADEVs reduce glioma cell chemokinesis and stimulus-induced cell invasion. Wound healing in GCM-ADEVs-treated GL261 was not modified with respect to the vehicle-treated cells (Appendix A), demonstrating that GCM-ADEVs do not affect the migration rate of glioma cells.

The ability of cells to move in response to a stimulus [30] is correlated to their ability to change shape and volume. The regulatory volume decrease (RVD) is a crucial process responsible for the re-establishment of the original volume after cell swelling. To investigate whether ADEV treatment could affect the ability of glioma cells to reorganize their morphology, we performed video-imaging experiments specifically designed to assess RVD during exposure to osmotic challenge. After the application of the 30% hypotonic solution, both GL261 cells were treated with the vehicle (a sEV-free fraction medium), and ADEVs underwent a rapid and marked increase of cell volume (swelling, 30–35% of increment), which reached a peak after 6 min from the beginning of the hypotonic solution exposure. Then, both cell groups started the active process of RVD. However, while vehicle-treated cells exhibited a rapid decrease of cell volume to reach a final cell volume even lower than the original one, ADEVs-treated GL261 cells displayed a markedly impaired RVD (Figure 2d). The quantitative analysis indeed showed that the percentage of RVD was 128.7 ± 9.1% and 62.6 ± 5.8% for vehicle- and ADEVs-treated cells, respectively (Figure 2d,e). This result confirmed the ability of ADEVs to modulate cell-shape change in glioma, even if the experimental system did not fully reflect the complex regulation of cell volume in the migration process. Of note, RVD in GCM-ADEVs-treated GL261 was not modified with respect to the vehicle-treated cells (Appendix A), demonstrating that GCM-ADEVs do not affect the cell volume of glioma cells.

### 3.3. ADEVs Reduce the I_Cl,swell_ Current and VRAC LRCC8A Subunit Expression in GL261 Cells

Since hypotonic stress is known to activate I_Cl,swell_ [31], we tested if ADEVs, shown to affect I_Cl,swell_-based RVD, do so by modulating I_Cl,swell_. To this end, we performed electrophysiological experiments using a conventional patch-clamp, whole-cell dialyzed configuration, in GL261 cells after 24 h of vehicle or ADEV treatment. I_Cl,swell_ density was assessed using current ramps from −100 to +50 mV from a holding potential of −40 mV and activating the current with a 30% extracellular hypotonic solution (Hypo). Upon Hypo application, a significant increase of the inward current was observed at −80 mV in both vehicle- and ADEVs-treated GL261 cells (Figure 3a,b). This hypotonic-activated current was fully blocked by 10 µM DCPIB, a potent and widely used selective blocker of I_Cl,swell_ (Figure 3a,b). The quantitative analysis of the DCPIB-sensitive current activated by the hypotonic stimulus showed that vehicle-treated cells exhibited a greater current density (41.9 ± 3.8 pA/pF) than ADEVs-treated cells (26.04 ± 3.6 pA/pF) (Figure 3c), indicating that ADEVs significantly downregulate I_Cl,swell_ in GL261 cells. This result was in line with the significant reduction of the RVD observed in ADEVs-treated cells (Figure 2d,e). Of note, the application of 10 µm DCPIB virtually abolished the re-acquisition of the original cell volume in GL261 cells, confirming the essential role of this current in the RVD process (Figure 3d,e).

I_Cl,swell_ is mediated by the voltage-regulated anion channel, VRAC, which is formed by heteromers of LRRC8 proteins, with LRRC8A being the only essential subunit which combines with at least another LRRC8 member (B–C-D-E) [32,33]. We verified if ADEV treatment modifies the expression of VRAC. To this end, we measured the mRNA expression of the LRRC8A subunit by qPCR. GL261 cells treated with ADEVs exhibited a significant reduction in the LRRC8A expression with respect to vehicle-treated cells (Figure 3f), indicating that the reduction of I_Cl,swell_ was mainly due to the significantly lowered VRAC gene expression.

### 3.4. ADEVs Downregulate VRAC LRCC8C Subunit Expression through miR124

The results obtained by q-PCR indicate an ADEVs-mediated interfering action with the VRAC gene expression. This would suggest that ADEVs might deliver genetic material, such as miRNAs, capable of interfering with the VRAC expression. A possible candidate would be miR124 as: (i) it is abundantly expressed in the CNS [34]; (ii) it exhibits a tumor suppressor role [34,35,36] (its silencing or downregulation promotes tumor malignancy); (iii) microglia-derived sEVs, highly enriched of miR124, exert an anti-glioma effect in mice [29].

This hypothesis is further supported by the observation that miR124 is highly expressed in ADEVs. In fact, the analysis of q-PCR showed that miR124 in ADEVs was 21 times more abundant than in microglia-derived sEVs (Figure 4a). Since ADEVs modify the expression of VRACs (Figure 3), we checked for a possible correlation between miR124 and the VRAC expression using a repository with experimentally supported miRNA–gene interactions. These computational analyses identified LRRC8C among miR124-validated target genes [37]. To confirm that LRRC8C is a direct target of miR-124, we generated a luciferase reporter construct containing a region of the 3′-UTR of LRRC8C (pmiR8C) which could bind miR124 (Figure 4b). We co-transfected GL261 cells with pmiR8C and a mimic sequence of miR124. The result showed that luciferase activity was decreased by miR124, suggesting a direct interaction between miR124 and LRRC8C-3′UTR (Figure 4c). We also observed a significant reduction of the LRRC8C gene and protein expression in GL261 transfected with the sequence mimic of miR124 compared to mock-transfected cells (Figure 4d–f). Moreover, a significant reduction of LRRC8C gene was observed in the GL261 treated with ADEVs with respect to vehicle-treated cells (Figure 4g). All these data indicate that ADEVs downregulate the transcription of the VRAC LRRC8C subunit through miR124.

To test whether miR124 would affect VRAC function, and thus the ability to modulate cell volume and invasion, we evaluated RVD, cell invasion and I_Cl,swell_ density, in GL261 cells transfected with the miR124 mimic sequence. Similar to what was observed for ADEV treatment, the overexpression of miR124 in GL261 cells: (i) impaired both RVD (Figure 5a,b) and cell invasion (Figure 5c) and (ii) significantly reduced the hypotonic-induced I_Cl,swell_ density (Figure 5d). Moreover, we performed immunofluorescence analysis in the tumor mass of ADEVs-treated mice; we observed a significant decrease in the LRRC8C protein expression with respect to vehicle-treated mice. This data confirms that in vivo ADEVs also reduce the VRAC LRRC8C expression. Altogether, these findings support the hypothesis that miR124 is one of the molecules transferred by ADEVs to GL261 cells that mediates vesicles anti-glioma effect downregulating VRAC.

## 4. Discussion

Astrocytes contribute to brain homeostasis regulating neuronal metabolism, maintaining the blood brain barrier (BBB) and communicating directly with other glial types, such as microglia and oligodendrocytes. Cell-to-cell contact is not always necessary because cells also communicate through the release of proteins (such as glypican 4, glypican 6 [38], thrombospondins [39,40], TGFβ [41], fibulin-2 [42], vimentin [43], Apolipoprotein D, FGF2, VEGF [42]) and miRNAs [43]. Recent studies demonstrated that most of these molecules [44,45,46] are transported to the target cells through EVs and that the transfer of genetic material (DNA and RNA) can induce epigenetic modifications on target cells [47].

Here, we demonstrate that ADEVs hinder glioma expansion, reducing both tumor volume and tumor cell proliferation in vivo and prolonging glioma-bearing mice survival. These effects reside in ADEV’s ability to impair both cell volume regulation (i.e., RVD) and cell migration and invasion in GL261 glioma cells by reducing the expression of VRAC. We demonstrated that the ADEV effect is mediated by miR124. In fact, the overexpression of miR124 in GL261 cells induced a strong reduction of migration and invasion and a significant impairment of cells to activate VRAC-dependent RVD. In the context of brain tumor, as well as in all the alterations of brain homeostasis [48], astrocytes acutely become reactive to promote the restoration of brain functionality and to fight the growth of cancer cells [49,50]. After some time, however, astrocytes become the target of tumor-released factors [51] which modify astrocytes towards a pro-tumoral program, for example maintained by the signal transducer and activator of transcription 3, STAT3 [52]. Astrocytes transform to support cancer survival, for example through the release of PEA-15 (the phosphoprotein enriched in astrocytes of 15 kDa [53]) or CCL20 (the chemokine C–C motif ligand 20 [54]), and cancer invasion through the release of cytokines such as TNF- α [55], IL-6 and IFN-γ, through the activation of the nuclear factor kappa B (NF-κB) and the transforming growth factor-β (TGF-β) signaling pathways [56]. Additionally, tumor-modified astrocytes promote an immunosuppressive pro-tumoral environment through the release of EVs that induce, in cancer cells, the loss of the tumor suppressor PTEN (Phosphatase and Tensin Homolog); this event causes an increased secretion of the chemokine CCL2, which recruits IBA1+ myeloid cells that strengthen brain metastatic tumor cell growth [13]. Our data are in line with these findings since EVs released by glioma-modified astrocytes increase the survival rate of GL261 cells, while EVs released by healthy astrocytes reduce the survival, the wound activity and the invasion of glioma cells.

Some recent reports indicated that EVs released from astrocytes [57,58] contribute to maintain the brain homeostasis through the transportation of specific miRNAs that activate microglia and leukocytes, promoting central and peripheral immune responses. Among many other factors carried by this type of vesicles, in search of the mechanisms underlying the anti-glioma effect of ADEVs, we have highlighted a presence of an already identified antitumor factor, which is miR124. The inhibition of miR124 in cancer cells promotes their growth and proliferation (for example in cervical carcinoma) as well as their invasion and migration (in gastric and non-small cell lung cancers [59,60]). In addition, miR124 overexpression reduced the cell vitality, proliferation and migration of breast cancer cells targeting flotillin-1 [61] (a protein of the non-caveolar lipid raft), correlated with the development of several tumors [62,63], or targeting SOX9 (sex-determining region Y-box9 protein) that is essential for self-renewal of tumor stem-like cells [64] in glioblastoma. Moreover, miR124 specifically reduced breast cancer cell invasion by targeting the transcription factor activating the enhancer-binding protein 4 V [65] and cadherin 2 [66]; in addition, miR124 reduced gastric cancer cell invasion by inhibiting the gene coding for another important protein for cell movement that is the integrin beta-3 [67]. Previous data from our group had demonstrated an enrichment of miR124 in small extracellular vesicles derived from microglia and that miR124 was involved in the anti-glioma effect exerted by these vesicles in mice [29]. Here, the enrichment of miR124 found in astrocyte vs. microglia derived sEVs encouraged the idea that miR124 could also play a key role in the ADEV antitumor process. The reduced ability of miR124-overexpressing glioma cells to invade (regardless of the presence of pro-invasive stimuli, Figure 4) confirmed our hypothesis, in line with all the already cited evidence [59,60].

All malignant features of GBM cells are interconnected acting co-operatively to create a TME that supports the survival and progression of tumor cells. Cell volume changes are indispensable for proliferation, migration and invasion processes. Glioma cells present abnormal proliferation and an elevated rate of migration and invasion that [68] contribute to the tumor-growth process and pose a huge obstacle to the development of drugs against this type of tumor. In fact, the high index of growth and the high capacity of invasion allow GBM cells to migrate from the primary site to different regions in brain parenchyma, reducing the efficacy of current therapeutical strategies [69,70]. Ion and water movement regulate cell volume, and thus cell growth, migration and invasion [71]. Our data demonstrated significant modulation of these cell functions in GL261 treated with ADEVs (Figure 3). All this evidence prompted us to hypothesize a link between the antitumor effect of ADEVs and the volume-regulated anion channels (VRAC). We demonstrated that ADEVs reduce I_Cl,swell_ and the gene expression of LRRC8A (Figure 3), that is the subunit essential for VRAC functionality [33]. Bioinformatics performed on the five different paralogs of LRRC8 (LRRC8A-E) predicted the highest probability of interaction between miR124 and LRRC8C [37]. We confirmed this suggestion by identifying a direct interaction between a 3′UTR region of LRRC8C and miR124 (Figure 4); moreover, we observed a significant reduced expression of the LRRC8C gene and reduced I_Cl,swell_ in cells overexpressing miR124 (Figure 4). The protein expression analysis on GL261 cells and on glioma-bearing mice confirmed a downregulation of the LRRC8C VRAC subunit upon ADEVs treatment (Figure 4 and Figure 5). All these findings indicate that the effects of ADEVs on I_Cl,swell_ are certainly partly dependent on miR124 transfer. Moreover, our results do not define whether the miR124-mediated downregulation of the LRRC8C gene has a direct or indirect effect on I_Cl,swell_. It has been recently demonstrated that in T cells, the LRRC8C subunit is essential for VRAC functionality, suggesting that the interaction between miR124 and LRRC8C may directly contribute to the modulation of I_Cl,swell_ also in glioma cells.

We demonstrated that EVs released by healthy astrocytes oppose GBM growth and invasion. One of the mechanisms of this effect is mediated by miR124 enriched in ADEVs, that reduces the expression of VRAC, one of the multiple regulators of cell shape and migration. VRACs are permeable also to large anionic excitatory amino acids (EAA) such as glutamate [72], and the specific VRAC blocker, DCPIB, inhibited the release of this amino acid contributing to the reduction of brain damage in ischemic stroke [73]. Since extracellular EAA concentrations increase in brain tumors (i.e., helping malignant cells to infiltrate by killing neurons), we could postulate that the reduction of the VRAC expression and function induced by ADEVs in glioma cells may explain the reduction of tumor growth observed in mice (Figure 1).

## Figures and Tables

**Figure 1 biomedicines-10-02952-f001:**
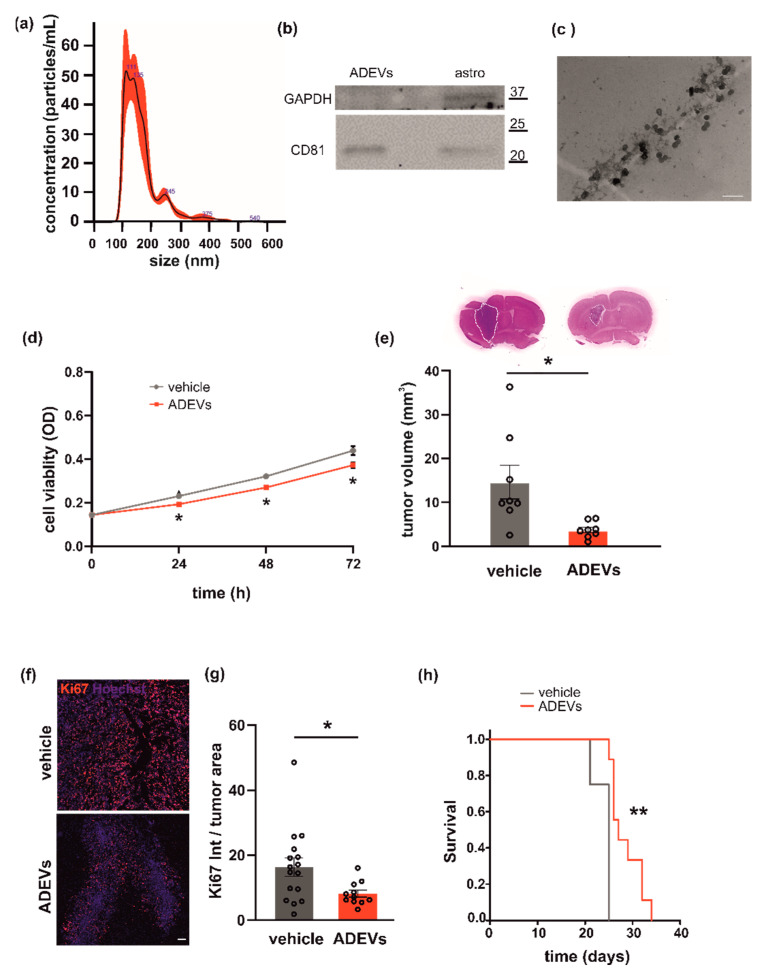
Analysis of astrocyte-derived sEVs (ADEVs) and their effect on GL261 morphology, vitality and on mice survival. (**a**) Representative concentration/size graph from the nanoparticle tracking analysis (NTA) of ADEVs. The average concentration (mean ± standard error, SE; in particles/mL) and mean size (±SE; in nm) of ADEVs of an example of five independent experiments is shown. (**b**) Western blot on ADEVs and astrocyte cell lysates (astro) for the exosomal marker CD81; GAPDH was used as a negative control. On the right, molecular weights are indicated. (**c**) Transmission electron microscopy of ADEVs. Bar = 200 nm, direct magnification 91,700×. (**d**) GL261 cells were assayed for cell viability with a sEV-free fraction medium (vehicle), in the presence of ADEVs obtained from healthy astrocytes (ADEVs) by MTT analysis. Cells were analyzed 0, 24, 48 and 72 h after plating. Viability was reported in mean optical density (OD) ± SE, N = 4; Student’s *t*-test, * *p* < 0.05. (**e**) Tumor size in the brain of GL261-bearing mice treated with a vehicle or ADEVs. Tumor size (in mm^3^) is reported as mean ± SE, N = 7/experimental group, Student’s *t*-test, * *p* < 0.05. On the top, representative coronal brain sections of GL261-bearing mice treated as above, stained with hematoxylin-eosin; the tumor area in the dashed line. (**f**) Representative immunofluorescence analysis for Ki67 (in red; Hoechst in blue) of coronal brain sections of GL261-bearing mice treated with a vehicle and with ADEVs, scale bar 100 µm. (**g**) Quantification of data are expressed as mean intensity (Int) per tumor area ± SE; N = 3, at least 4 fields per condition; Student’s *t*-test, * *p* < 0.05. (**h**) Kaplan–Meier survival curves of GL261-bearing mice treated with ADEVs. N = 9 mice/experimental group; Gehan-Breslow statistic for the survival curves (** *p* < 0.001).

**Figure 2 biomedicines-10-02952-f002:**
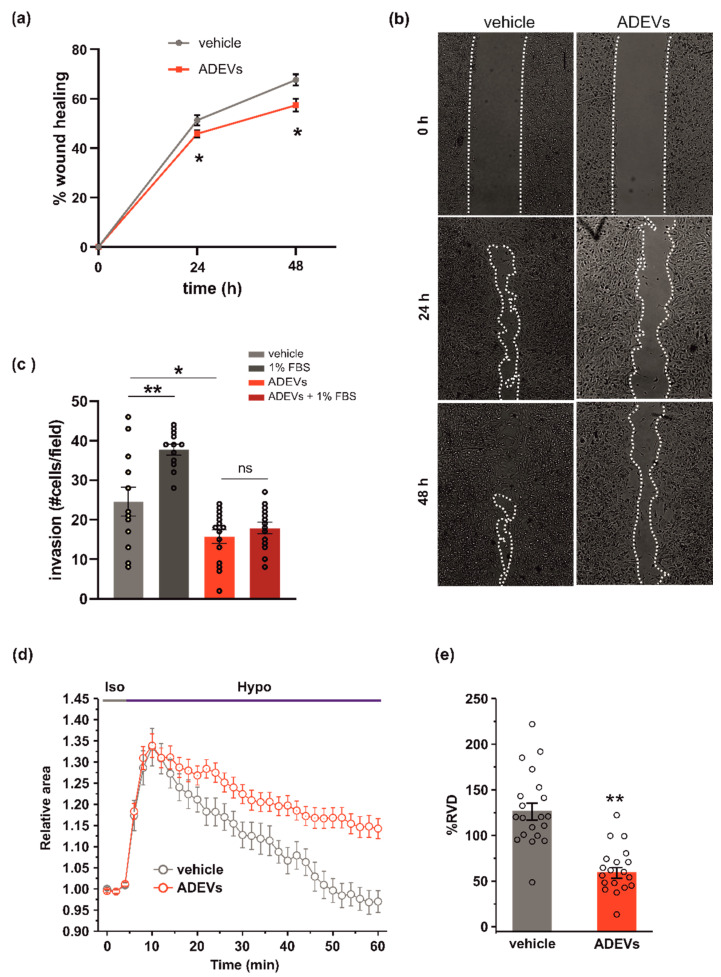
ADEVs limit glioma cell invasion. (**a**) GL261 glioma cells were treated with a sEV-free fraction medium (vehicle) or ADEVs, and a wound healing (migration) assay was performed. GL261 migration was measured 24 and 48 h after treatment, data are expressed as a mean percentage of wound healing area ± SE, N = 4, Student’s *t*-test, * *p* < 0.001. (**b**) Representative images of wound healing assay on GL261 cells treated as in (**a**). (**c**) GL261 cells were assayed for basal or FBS-induced invasion in vehicle- or ADEVs-treated cells on Matrigel. GL261 invasion was measured as a mean of invading cells/field ± SE, N = 3, at least 4 fields/condition; Student’s *t*-test, * *p* < 0.05 and ** *p* < 0.001 (**d**) Time course of RVD as evaluated from the changes of the relative cell area (Arel) in vehicle-treated cells (Iso, grey bar) and during the application of a 30% hypotonic solution (Hypo, blue bar) to GL261 vehicle-treated cells (vehicle, grey circles, *n* = 20) and to ADEVs-treated GL261 cells (ADEVs, red circles, *n* = 19). (**e**) Bar plot showing the average percentage of RVD in vehicle- and ADEVs-treated GL261 cells, calculated as follows: % RVD = (Arel,peak−Arel,56 min)/(Arel,peak)) × 100. Data are shown as mean ± SEM. Student’s *t*-test, ** *p* < 0.001.

**Figure 3 biomedicines-10-02952-f003:**
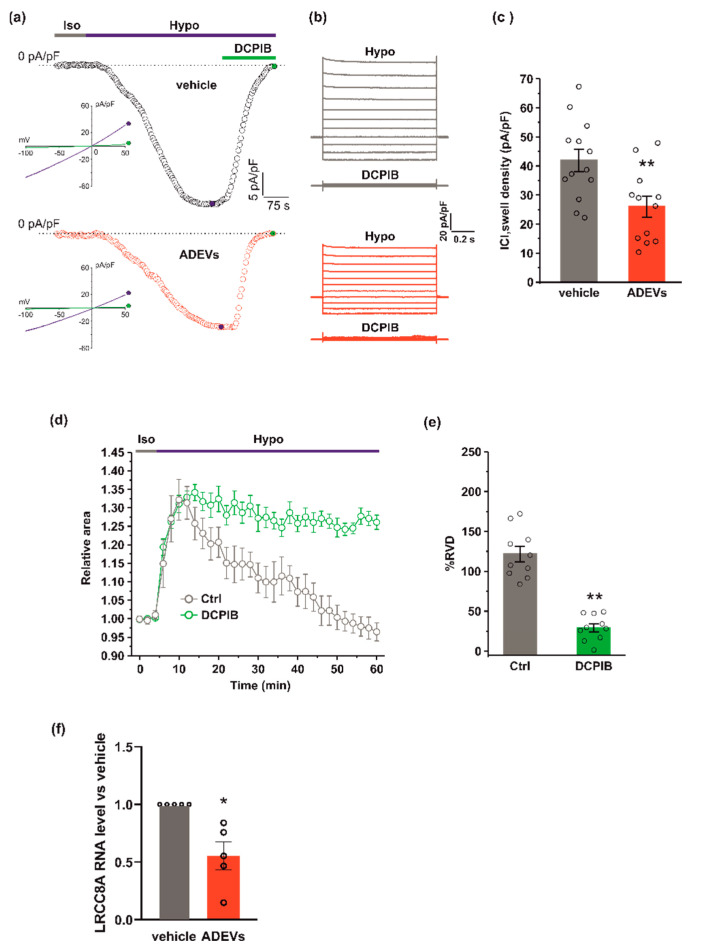
ADEVs reduce I_Cl,swell_ and the LRCC8A expression in GL261 cells.(**a**) Representative time courses of I_Cl,swell_ density (pA/pF) measured from current ramps at −80 mV, the equilibrium potential for K+ under our recording conditions, during the application of a 30% hypotonic solution (Hypo, blue bar) and after the addition of 10 µM DCPIB (green bar), in vehicle-treated GL261 cells (vehicle, black circles, top) and in ADEVs-treated GL261 cells (ADEVs, red circles, bottom). Insets: representative current ramps from −100 to 50 mV (600 ms duration) from a holding potential of −40 mV, under a 30% hypotonic solution (blue trace) and a 30% hypotonic solution containing 10 µM DCPIB (green trace). (**b**) Representative families of current traces evoked by applying 1 s voltage steps from −100 to 120 mV, in steps of 20 mV, from a holding potential of - 40 mV in the presence of a 30 %hypotonic solution, and in the presence of a hypotonic solution containing 10 µM DCPIB, in vehicle-treated (grey traces, top) and in ADEVs-treated (red traces, bottom) GL261 cells. (**c**) Bar plot showing the average current density measured at - 80 mV during exposure to a 30% hypotonic solution, in vehicle- (*n* = 13) and ADEVs-treated (*n* = 12) GL261 cells. Data are shown as a mean ± SEM; ** *p* < 0.01. (**d**) Time course of RVD as evaluated from the changes of the relative cell area in vehicle conditions (Iso, grey bar) and during the application of a 30% hypotonic solution (Hypo, blue bar) to vehicle-treated GL261 cells, either in the absence (Ctrl, grey circles, *n* = 20) or presence of 10 µM DCPIB (green circles, *n* = 10). (**e**) Bar plot showing the average percentage of RVD in GL261 cells in the absence (Ctrl) or presence of 10 µM DCPIB (DCPIB), calculated as follows: %RVD = (Arel,peak−Arel,56 min)/(Arel,peak)) × 100. Data mean ± SE; Student’s *t*-test ** *p* < 0.0001. (**f**) qPCR of LRCC8A gene in vehicle-treated GL261 cells (vehicle) or GL261 cells treated with ADEVs. Data are the mean ± SE of fold increase vs. vehicle, normalized on *gapdh* (used as a housekeeping gene); N = 5; * *p* < 0.05 vs. ctrl; by using the Mann–Whitney rank sum test.

**Figure 4 biomedicines-10-02952-f004:**
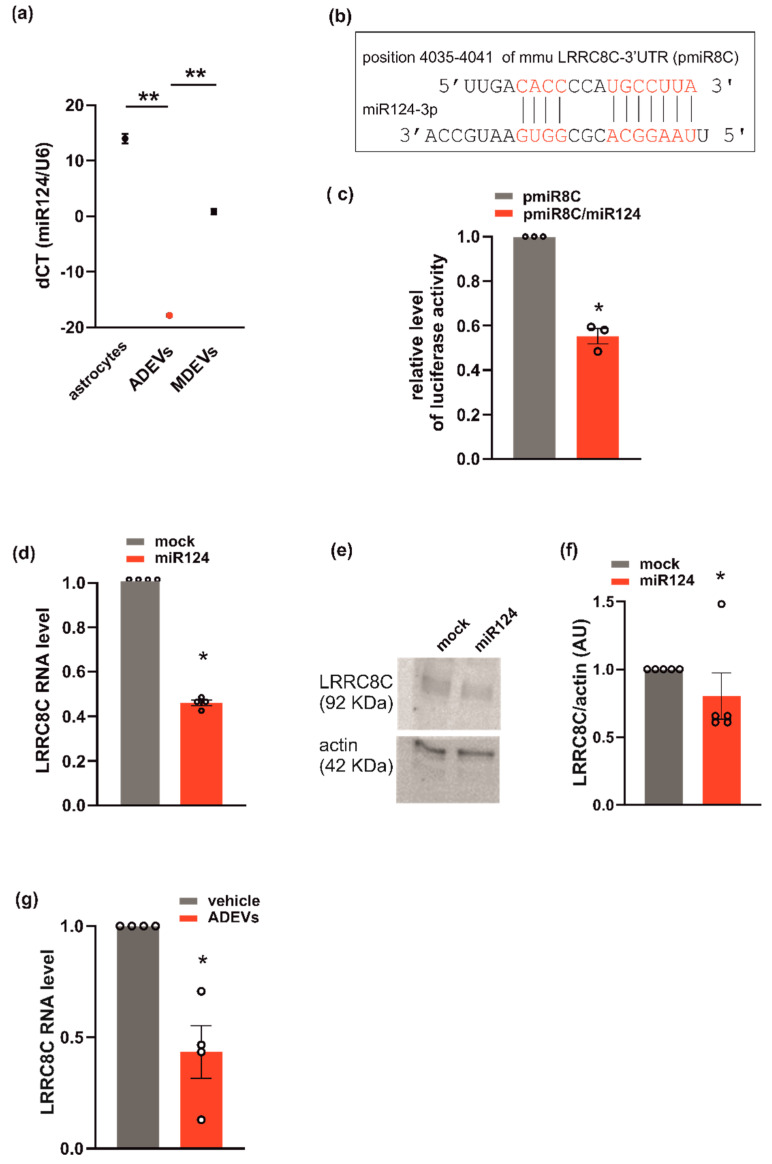
miR124 and LRRC8C VRAC subunit interaction in GL261. (**a**) miR124 gene expression in astrocytes, ADEVs and microglia-derived sEVs (MDEVs) were evaluated by q-PCR analysis. Data are expressed as the mean ± SE of ΔCt, normalized to U6 (used as a housekeeping gene); N = 3; Student’s *t*-test, ** *p* < 0.001; (**b**) miR124 binding site on 3′-UTR of LRRC8C is shown (pmiR8C, in red); (**c**) GL261 transfected with pmiR8C or co-transfected with pmiR8C and miR124 mimic sequence were analyzed for luciferase reporter gene assay. Data are the mean ± SE of fold change; N = 3; Student’s *t*-test, * *p* < 0.05. (**d**) q-PCR of LRCC8C gene in GL261 cells transfected with mock or miR124 mimic sequence. Data are the mean ± SE of fold increase vs. mock, normalized to U6 (used as a housekeeping gene); N = 4; Student’s *t*-test, * *p* < 0.05. (**e**) Western blot on lysates of GL261 cells transfected with mock or miR124 mimic sequence for LRRC8C; actin was used as a loading control. (**f**) Densitometry analyses of blots for LRRC8C was normalized on actin (arbitrary unit, AU). Data are the mean of AU ± SE, N = 5; Student’s *t*-test, * *p* < 0.05. (**g**) qPCR of LRCC8C gene in vehicle- or ADEVs-treated GL261 cells. Data are the mean ± SE of fold increase vs. vehicle, normalized on *gapdh* (used as a housekeeping gene); N = 4; Student’s *t*-test, * *p* < 0.05.

**Figure 5 biomedicines-10-02952-f005:**
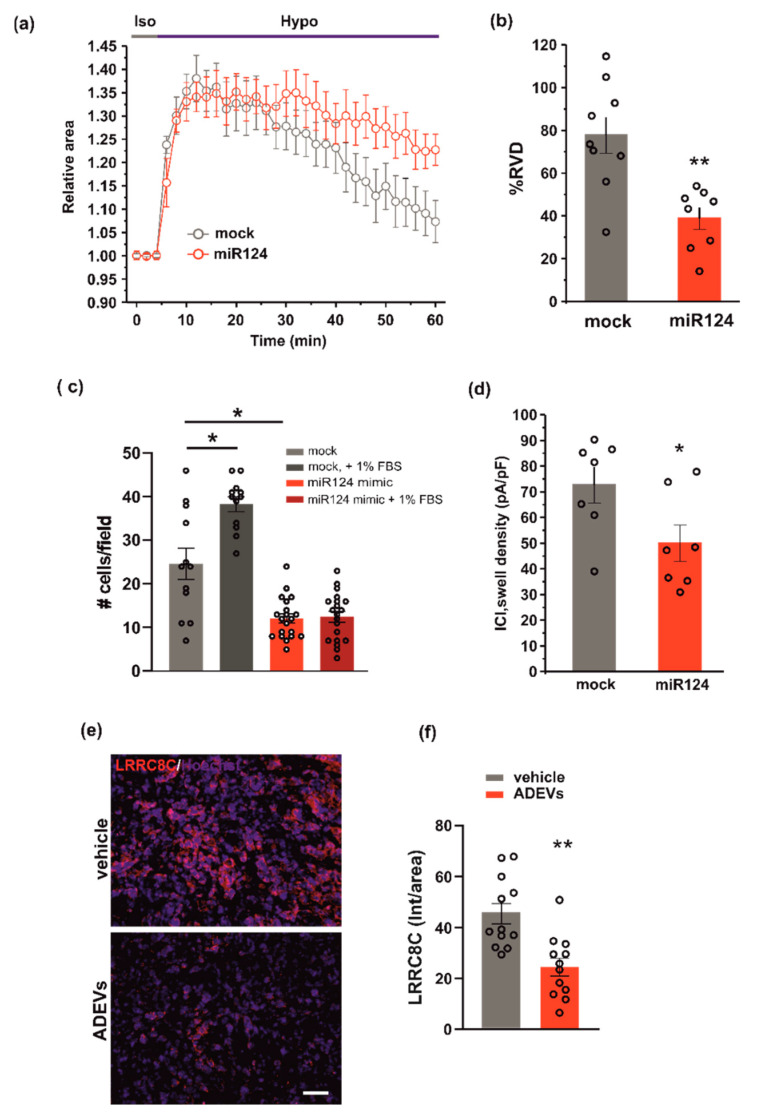
miR124 involvement in ADEVs-mediated effects on GL261 cells. (**a**) Time course of RVD as evaluated from the changes of the relative cell area in vehicle-treated cells (Iso, grey bar) and during the application of a 30% hypotonic solution (Hypo, blue bar) in control mock GL261 cells (mock, grey circles, *n* = 9) and in miR124-transfected (miR124, *n* = 12, red circles) GL261 cells. (**b**) Bar plot showing the average percentage of RVD in mock- and miR124-transfected GL261 cells, calculated as follows: %RVD = (Arel,peak−Arel,56 min)/(Arel,peak)) × 100. Data mean ± SE; Student’s *t*-test ** *p* < 0.01. (**c**) GL261 mock- or miR124-transfected GL261 cells were assayed for basal- or FBS-induced invasion on Matrigel. GL261 invasion was measured as a mean number of invading cells/field ± SE, N = 3, at least 4 fields per condition; Student’s *t*-test, * *p* < 0.05. (**d**) Bar plot showing the average current density measured at −80 mV during the exposure to a 30% hypotonic solution, in mock- (*n* = 13) and miR124-transfected (*n* = 12) GL261 cells. Data mean ± SE; Student’s *t*-test * *p* < 0.05. (**e**) Representative immunofluorescence analysis for LRRC8C (in red; Hoechst in blue) of coronal brain sections of GL261-bearing mice treated with a vehicle and with ADEVs, scale bar 50 µm. (**f**) Data are expressed as a mean intensity (Int) per area ± SE; N = 3, 4 fields per condition; Student’s *t*-test, * *p* < 0.05.

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
