# Peer review of "Astrocytes-Derived Small Extracellular Vesicles Hinder Glioma Growth"

_biomedicines, 2022, doi:10.3390/biomedicines10112952_

Round 1

Reviewer 1 Report

Serpe and colleagues have written an article about astrocyte-derived small EVs exert a defensive mechanism against tumour cell growth and invasion using murine glioma GL261 cells as model. Although the topic is very interesting, the experiments are not properly planned, the conclusions are not well sustained and therefore the article is not ready for publication.

In general, experiments have been done comparing exosomes with probably just a chemical vehicle (i.e PBS, although not clear). Therefore, it is normal that sEV that contains RNA, DNA and proteins have more effect the just PBS. For the same reason it is just expected that if you extract sEV from tumour cells or GCM ADEVs, there will be a protumoral effect (even if the authors just did couple of experiments in this direction). It would be interesting to see if non-tumoral sEV have not such an effect (i.e. Microglia derived sEV, that they already have and have shown to have less miR124).

Altogether the article seems unfinish, and not well sustained.

Some Major concerns:

- WB characterization of the sEV would be desirable.

- Figures should be located next to the place where they are cited and not as an independent section.

- It looks like when the authors state “vehicle” that only means the absence of ADEVs (line 385). - Is it just PBS? That is not the right control for this experiment. 

- It is not specified the number of cells they have implanted. They have to be sure they have put the same, because differences in tumour size could be due to a different number of cells being implanted. The clean way to do that is to follow the tumour growth in the same mice by cranial tomography and see a reduction after injection.

-It is not clear, why GCM ADEVs experiment disappear from the in vivo experiments, and from the rest of the article.

- The regulatory volume decrease experiments are not convincing.

- Since the entire point of looking for RVD and ICI,swell is the fact that it is supposed to happened in GBM. It would be interesting to see the effect in vivo. Since they already have the system, they could patch-clamp the cells inside the tumour in a brain slide and compare with the treated situation.

Some small concerns:

- Typo mistakes (i.e. Line 59, has empty brackets).

- Extra final points. (i.e, line 547, line 107)

- The font size makes unreadable the graphical abstract.

Reviewer 2 Report

Extracellular vesicles (EVs) are membrane-contained vesicles enclosing proteins and distinct species of RNA released by cells into the microenvironment. In glioma, studies have focused on the the role of EVs on glioma progression and find new diagnostic biomarker in EVs.  Here, Serpe et al, submitted a manuscript found that astrocyte-derived small EVs inhibits glioma proliferation and invasion and reveal that miR124 in EVs decrease LRRC8C expression to affect Volume-regulated Anion Channels (VRAC) function. Therefore, the manuscript has its merit. A list of my comments is given below.

1.    The authors found that ADEVs decreases glioma cell proliferation in Figure 3C. What’s the concentration did authors use here? More concentrations are need to double confirm the role of ADEVs.

2.    Could the authors explain why the astrocytes generate EVs in GCM which can promote glioma cell proliferation?

3.    As the authors mentioned in figure 1d, the authors treated the mice with ADEVs after implantation for 7 and 14 days. But, from figures 1d-1g, there’s one ADEVs group. How many days did the authors treat the mice? And, what’s going on of another ADEVs group?

4.    Can the authors prove or hypothesize the mechanics of ADEVs block the role of FBS on invasion?

5.    The western blot is needed to confirm the roles of miR124 and ADEVs on LRRC8C expression.

Round 2

Reviewer 1 Report

Serpe and colleagues have answer properly some of the presented issues, mostly the ones related with editing the manuscript. However, the major concerns have not been solved and the offered explanations are not sufficient to avoid doing the experiments. It is important because there are not major conclusions in the article.

Those major concerns still stand and there are some other minor points:

- Low quality of the Graphical Abstract. Stil unreadable.

- Low quality, and small size of the pictures in figure 1. Specially an issue in e and f.

- The change in line 22o of Matrigel> atrigel-coated transwells. It makes no sense.

-iScript Reverse Transcription Supermix.

- Line 325: astrocyte cell lysates.

- Fig. 2 should be rearrange so that panel b has more space. The contrast should be adjusted so that we can distinguish something.

Reviewer 2 Report

The authors addressed most of my concerns. However, the authors gave the conclusion of ADEVs regulates LRRC8C to affect glioma cells phenotype based on the PCR results which is not that convinced. The role of ADEVs on LRRC8C expression is one of the most essential parts to support the conclusion. If applicable, the author should check the LRRC8C expression using different methods. If applicable, the author should knock down the LRRC8C to test whether the role of ADEVs can be inhibited.

Round 3

Reviewer 1 Report

The article can be accepted in the present form, however it would be nice if the authors improve the resolution of the pictures (i.e. Fig 1f ).

Reviewer 2 Report

Thank for addressing my concerns.